# The Degradation of Botulinum Neurotoxin Light Chains Using PROTACs

**DOI:** 10.3390/ijms25137472

**Published:** 2024-07-08

**Authors:** Yien Che Tsai, Loren Kozar, Zo P. Mawi, Konstantin Ichtchenko, Charles B. Shoemaker, Patrick M. McNutt, Allan M. Weissman

**Affiliations:** 1Cancer Innovation Laboratory, Center for Cancer Research, National Cancer Institute, Frederick, MD 21702, USAweissmaa@mail.nih.gov (A.M.W.); 2Department of Biochemistry and Molecular Pharmacology, New York University Grossman School of Medicine, New York, NY 10016, USA; konstantin.ichtchenko@nyulangone.org; 3Department of Infectious Diseases and Global Health, Tufts University Cummings School of Veterinary Medicine, Grafton, MA 01536, USA; charles.shoemaker@tufts.edu; 4Wake Forest Research Institute for Regenerative Medicine, Wake Forest University School of Medicine, Winston-Salem, NC 27101, USA; pmcnutt@wakehealth.edu; 5Women’s Malignancies Branch, Center for Cancer Research, National Cancer Institute, Bethesda, MD 20892, USA

**Keywords:** toxin persistence, PROTAC, ubiquitin–proteasome system, synthetic toxins, persistence, deubiquitinating enzymes, iatrogenic botulism

## Abstract

Botulinum neurotoxins are some of the most potent natural toxins known; they cause flaccid paralysis by inhibiting synaptic vesicle release. Some serotypes, notably serotype A and B, can cause persistent paralysis lasting for several months. Because of their potency and persistence, botulinum neurotoxins are now used to manage several clinical conditions, and there is interest in expanding their clinical applications using engineered toxins with novel substrate specificities. It will also be beneficial to engineer toxins with tunable persistence. We have investigated the potential use of small-molecule proteolysis-targeting chimeras (PROTACs) to vary the persistence of modified recombinant botulinum neurotoxins. We also describe a complementary approach that has potential relevance for botulism treatment. This second approach uses a camelid heavy chain antibody directed against botulinum neurotoxin that is modified to bind the PROTAC. These strategies provide proof of principle for the use of two different approaches to fine tune the persistence of botulinum neurotoxins by selectively targeting their catalytic light chains for proteasomal degradation.

## 1. Introduction

Botulinum neurotoxins (BoNTs) cause flaccid paralysis by inhibiting synaptic transmission [1,2]. At least seven serotypes of BoNT have been described, all of which cleave one or more of the SNARE (soluble N-ethylmaleimide-sensitive factor attachment protein receptor) proteins [3,4]. BoNTs consist of a heavy chain (HC), which binds to cell surface receptors, and a catalytic light chain (LC) that, upon cell entry, specifically cleaves one of the SNAREs thereby inhibiting synaptic vesicle release [5,6]. LCs of the different BoNT serotypes target different SNARE proteins: serotypes A and E LCs cleave 25 kDa synaptosomal-associated protein (SNAP25); serotypes B, D, F, and G LCs cleave synaptobrevin; and serotype C LC cleaves SNAP25 and syntaxin-1 [4,7]. These toxins show exquisite substrate specificity, as illustrated by the fact that BoNT/A cleaves SNAP25 after residue 197 and BoNT/E LC cleaves SNAP25 after residue 180. The different BoNT serotypes cause paralysis of different durations because of differences in toxin persistence [8]. Serotypes A and B are characterized by long persistence, producing neuromuscular paralysis that can last several months [9]. This unique combination of high avidity for motor neuron presynaptic receptors, specific targeting of residues essential for neurotransmitter release, and long-lasting paralysis underlies the extraordinary toxicity of these BoNTs and has also made these molecules valuable clinical tools for treating numerous neuromuscular indications [10,11].

Therapeutic BoNT injections are the first-line treatment for conditions including cervical dystonia, blepharospasm, and focal spasticity, as well as overactive bladder and several autonomic disorders [10]. However, the persistence of BoNT serotypes A and B also results in long-lasting adverse effects following excessive or mis-localized toxin injections [12]. Iatrogenic botulism is most common with therapeutic treatments involving relatively high dose injections or injections of deep muscle groups in, for example, treatment of lower limb spasticity, dystonia, or cerebral palsy [13,14,15]. Iatrogenic botulism symptoms can range from local effects, such as ptosis, diplopia, or dysphagia, to systemic effects, such as generalized weakness and respiratory distress and can result in death. Although several studies have recently described potential botulism countermeasures, there is currently no clinical treatment for post-symptomatic botulism [16,17,18,19,20].

The development of methods to promote BoNT clearance may present a novel treatment strategy for iatrogenic botulism. We previously found that differences in persistence of serotypes A and E are attributable to differences in LC engagement with the cellular protein degradation machinery [21,22,23,24]. Specifically, BoNT/E LC is ubiquitinated by the ubiquitin ligase (E3) TRAF2 and degraded rapidly [22]. Conversely, although BoNT/A LC is ubiquitinated by a different E3, HECTD2, it is not degraded due to the activity of at least two de-ubiquitinating enzymes (DUBs) that counteract HECTD2 ubiquitination [21]. Because the turnover of BoNT/A LC depends on a dynamic balance of ubiquitination and de-ubiquitination, we hypothesize that manipulating the ubiquitin–proteasome system (UPS) to enhance LC ubiquitination will shorten BoNT/A persistence.

Previous efforts to modulate the rate of turnover of BoNT LC in cells were based on designer E3 ligases that target BoNT/A LC to the UPS [22,25]. In one design, we fused an E3 domain to a SNAP-25 that is resistant to BoNT/A cleavage [22]. The SNAP-25 mutant binds BoNT/A LC allowing the E3 domain to conjugate ubiquitin on the toxin leading to its proteasomal degradation. This chimeric SNAP-25 mutant is useful as proof of concept but is relatively large, making cell entry problematic. In another design, we fused a minimal F-box derived from β-TrCP to a BoNT/A LC-specific nanobody, VHH B8, which is an antigen-binding domain derived from a camelid heavy-chain-only antibody [26]. VHH B8 binds BoNT/A LC with nanomolar affinity while the minimal F-box recruits other components of the Skp1–Cullin1–F-box (SCF) RING ligase family, which is part of the cullin RING ligase (CRL) superfamily, to target BoNT/A LC for ubiquitination and degradation [25]. This VHH–F-box chimera has the additional advantage that VHH B8 also inhibits the catalytic activity of BoNT/A LC [26]. However, the F-box renders the chimera itself a target for proteasomal degradation. Thus, while these constructs successfully promoted degradation of intracellular toxin, they had low potency for reasons that include poor cellular uptake and short intracellular persistence. Moreover, these constructs did not allow for the ability to precisely control the timing and rate of degradation of BoNT/A LC.

Here, we describe a next generation of BoNT persistence modulators based on the use of proteolysis-targeting chimeras (PROTACs) to accelerate clearance of BoNT/A [27]. PROTACs are bifunctional molecules composed of a ligand that binds the protein of interest linked to a ligand that binds a specific E3 ubiquitin ligase. The linker provides a suitable molecular distance and orientation for simultaneous binding of the protein of interest and the E3 ligase in a ternary complex, thus promoting ubiquitination and degradation of the target protein. Early PROTACs used peptides derived from Hypoxia-Inducible Factor-1α (HIF-1α) to recruit the Von Hippel–Lindau (VHL) CRL complex [27]. The discovery of small-molecule mimetics of a HIF-1α peptide made possible the design of cell-permeant, small-molecule PROTACs [28,29]. Recent advances have made available PROTACs that recruit different E3s including the Cereblon (CBRN) CRL complex, and Parkin [30,31,32]. At the same time, ligands for various targeting domains including HaloTag and a mutant of 12-KDa FK506-binding protein (FKBP12) have been developed [33,34].

As a proof-of-concept study to assess the feasibility of this approach, we undertook two methods to evaluate the ability of PROTACs to promote BoNT/A clearance. First, we engineered recombinant BoNT/A LCs fused to peptide sequences that are targeted by commercially available PROTACs. These studies confirmed that existing E3 ligase pathways could be exploited to modulate LC persistence. Next, we examined the ability of PROTACs to promote clearance of wild-type BoNT/A LC using E3 recruitment domains directly fused to a BoNT/A-specific VHH. In this way we tested two methods that increase the repertoire of ligands and E3s available for modulating BoNT persistence. These studies showed that BoNT LC persistence can be regulated by PROTAC treatment, opening the door for the development of PROTACs that directly bind to BoNT LCs to target and ubiquitinate wild-type BoNTs.

## 2. Results and Discussion

### 2.1. BoNT Persistence in Animals Correlates with the Rate of Protease Turnover in Transfected Cells

Several studies suggest that the rate of ubiquitin-dependent, proteasome-mediated degradation of the light chain is a primary determinant of toxin persistence [21,22,24,25]. Specifically, the rapid turnover of BoNT/E LC correlates with its short persistence in vivo [22]. To test the effects of the UPS on the persistence of BoNT/A subtypes, we first screened three BoNT/A subtypes (A1–A3) for differences in intracellular LC stability. To compare the relative stabilities of each subtype, we used an established stability assay in which HEK-293 cells are transfected with plasmids encoding GFP-LC/A fusion proteins for 36 h followed by immunoblot evaluation of GFP-LC/A levels at 0, 3, and 6 h of treatment with 50 μg/mL cycloheximide (CHX), to inhibit new protein synthesis. It is well-established that the levels of different GFP-fusion proteins can be reliably monitored using GFP fluorescence or immunoreactivity [21,22,25,35]. Of the three subtypes tested, LC/A1 and LC/A2 did not undergo detectable degradation during 6 h of cycloheximide treatment, whereas LC/A3 levels were reduced by 70% (Figure 1A). This is consistent with previous studies showing that BoNT/A3 exhibits the shortest duration of paralysis in vivo [36] and the shortest duration of SNAP25 cleavage in cultured neurons [37]. In contrast, BoNT/A1 and A2 exhibit prolonged toxin persistence in vitro and in vivo [36,37].

We have previously determined that DUBs USP9X and VCIP135 were important for BoNT/A LC persistence in cells [21]. As there is currently no selective inhibitor for VCIP135, we evaluated the effects of WP1130, a selective USP9X inhibitor, on LC/A1 stability. WP1130 increased the rate of degradation of LC/A1 whereas a selective USP7 inhibitor, P22077, had no apparent effect on LC/A1 turnover (Figure 1B). These results suggest that modulation of the UPS system using DUB inhibitors may be effective for reducing BoNT/A persistence. Consistent with these results, Kiris and colleagues found in a recent screen for small-molecule modulators of BoNT/A toxicity that WP1130 and PR619, a broad-spectrum DUB inhibitor, protected SNAP25 from proteolysis by BoNT/A [38]. However, because USP9X is critical for cell survival [39], it may be difficult to accelerate BoNT/A LC degradation without incurring severe adverse effects. 

### 2.2. Targeting BoNT/A Light Chain with PROTAC

Another strategy to alter BoNT/A persistence is to enhance ubiquitination of BoNT/A LC in cells. We have previously found that chimeric molecules fused to E3 ligase domains can promote ubiquitination and clearance of BoNT/A LC [21,22,25]. There is emerging interest in designing proteolysis-targeting chimeras (PROTACs) to degrade target proteins of interest. One major challenge in PROTAC development is the availability of a ligand that specifically binds the target protein. Since BoNTs can be produced recombinantly, a more versatile approach is to fuse well-established targeting domains to the BoNT LC.

As initial proof of concept, we fused a HaloTag to GFP-LC/A1 (Figure 2A). HaloTag is a haloalkane dehalogenase mutated to covalently bind to a synthetic ligand [40]. Several HaloPROTACs have been developed that bind the HaloTag and recruit an E3 to increase the ubiquitination of target proteins and thereby enhancing their degradation [33,41]. We choose HaloPROTAC3 as it can induce complete loss of several proteins fused to HaloTag at 1 μM and is not cytotoxic up to 10 μM [33], as measured by mitochondrial function. To further assess the potential toxicity of HaloPROTAC3, we treated HEK-293 cells with different concentrations of HaloPROTAC3 for 24 h and directly assayed cell viability based on cell membrane integrity. Cell viability was minimally affected by treatment with HaloPROTAC3 up to 100 μM (Figure 2B). 

We then proceeded to test whether HaloPROTAC3, which recruits the VHL E3 complex, induces the degradation of HaloTag-LC/A1. We expressed GFP-Halo-LC/A1 in HEK-293 cells by plasmid transfection and treated the cells with a range of concentrations of HaloPROTAC3 for 24 h. HaloPROTAC3 reduced the levels of GFP-Halo-LC/A1 in a dose-dependent manner (Figure 2C). The effect was specific, as HaloPROTAC3 had no effect on GFP-LC/A1 lacking the HaloTag (Figure 2D). The concentration of HaloPROTAC3 producing half maximal degradation of GFP-Halo-LC/A1 (DC_50_) was ~8 μM (Figure 2E), which was relatively high compared to previously described targets [33]. To assess whether HaloPROTAC3 increased the ubiquitination of GFP-Halo-LC/A1 at those elevated doses, we expressed GFP-Halo-LC/A1 in HEK-293 cells by plasmid transfection and treated the cells with 20 μM of HaloPROTAC3 for 20 h. The cells were then treated for 4 h with 30 μM of MG132, which inhibits the proteasome, to accumulate ubiquitinated proteins. HaloPROTAC3 substantially reduced the levels of GFP-Halo-LC/A1 but did not alter the total cellular ubiquitination (Figure 2F, Right Panel). However, HaloPROTAC3 substantially increased the ubiquitination of GFP-Halo-LC/A1 (Figure 2F, Left Panel; compare Lanes 3 and 4), despite a decrease in the total level of GFP-Halo-LC/A1, consistent with the well-established mechanism of action for PROTACs.

The HaloTag peptide is relatively large at ~33 kDa, which limits its ability to be delivered to cells as part of a chimeric holotoxin. In addition, the DC_50_ of HaloPROTAC3 for HaloTag-LC/A1 is high compared to previously described targets [33]. For these reasons, we assessed an alternative, the ~11 kDa *m*FKBP tag, which is derived from mutant human FKBP12 (F36V). Several PROTAC molecules are available for the *m*FKBP tag, including dTAG^V^-1 and dTAG-13 which are VHL- and CBRN-binding ligands, respectively. Furthermore, these PROTACs specifically bind the F36V mutant of FKBP12 (*m*FKBP12) but not endogenous wild-type FKBP12, thus reducing the likelihood of off-target toxicities [34].

We expressed *m*FKBP-tagged LC/A1 (*m*FKBP-LC/A1) by plasmid transfection in HEK-293 cells and then treated the cells with increasing doses of dTAG^V^-1 (Figure 3A). As the concentration of dTAG^V^-1 increased, the steady state levels of *m*FKBP-LC/A1 decreased, reaching a low of ~1 μM. A further increase in dTAG^V^-1 concentration resulted in a rise in *m*FKBP-LC/A1 levels (Figure 3B). This biphasic dose response is consistent with the PROTAC “hook effect”, which results from the formation of dimeric complexes of PROTAC with the E3 or with LC/A1 at saturating concentrations [42,43]. Strikingly, dTAG^V^-1 reduced the level of *m*FKBP-LC/A1 with an apparent DC_50_ of ~5 nM (Figure 3B), a three orders of magnitude improvement over targeting with HaloPROTAC3. We next assessed the effect of dTAG^V^-1 on the turnover of *m*FKBP-LC/A1 in M17 neuroblastoma cells in the presence of cycloheximide. Treatment with dTAG^V^-1 increased the degradation of *m*FKBP-LC/A1 in M17 cells whereas treatment with an inert PROTAC, dTAG^V^-1-NEG, had no effect (Figure 3C). 

### 2.3. Targeting BoNT/A Light Chain Using a VHH and PROTAC

We next considered an alternative approach to targeting BoNT/A LC lacking a targeting domain, as most currently approved toxins for medical use and the wild-type toxins related to botulism intoxication are unmodified. In this situation, the targeting domain could be provided through an intermediary protein that binds BoNT LC with high affinity. For this, we used a camelid VHH that recognizes the specific BoNT LC. The camelid VHH B8 binds specifically to BoNT/A LC with ~1 nM affinity and inhibits its protease activity [26]. Furthermore, VHH B8 can be delivered in vivo using an atoxic derivative of BoNT/C1 to prolong survival of animals intoxicated with BoNT/A [18,44]. Therefore, we sought to design a VHH construct that can both inhibit LC protease activity and accelerate LC turnover.

In one design, we fused HaloTag to VHH B8, which also includes GFP and E-tag for detection. PROTACs normally function by forming a ternary complex with the target protein and E3 complex. However, we postulated that the high affinity of VHH B8 for BoNT/A LC would allow the PROTAC to function in a quaternary complex with B8 as an intermediary protein that would couple the PROTAC to E3 and BoNT/A LC (schematized in Figure 4A). To test this idea, we co-expressed HaloTag-B8 and GFP-LC/A1 in HEK-293 cells and then treated the cells with a range of doses of HaloPROTAC3 for 24 h. As might be expected, as the concentration of HaloPROTAC3 increased, the levels of its binding partner, and therefore direct target, HaloTag-B8, decreased (Figure 4B, Middle Panel). The response once again appeared biphasic, as levels of HaloTag-B8 were lowest at ~20 μM HaloPROTAC3 and increased when the concentration was raised to 40 μM (Figure 4B, Middle Panel; Lanes 8 and 9, respectively. More importantly, HaloPROTAC3 reduced the levels of GFP-LC/A1, achieving a DC_50_ for GFP-LC/A1 of ~4 μM (Figure 4B, Upper Panel; Figure 4C).

We next evaluated a fusion of the *m*FKBP domain to VHH B8. In cells expressing *m*FKBP-B8 and GFP-LC/A1, dTAG^V^-1 reduced the levels of GFP-LC/A1, with a DC_50_ of ~20 nM (Figure 5A,C). Similar to the direct fusion of *m*FKBP tag to BoNT/A LC, we again observed an almost three orders of magnitude improvement in targeting with dTAG^V^-1 compared to HaloPROTAC3. This effect was specific, as dTAG^V^-1 failed to lower GFP-LC/A1 levels in cells co-expressing VHH B8 without a *m*FKBP tag (Figure 5B,C). 

In medical cases of botulism poisoning, further uptake of the toxin can be blocked with neutralizing antibodies. Toxin persistence is caused by the pool of BoNT/A LC that entered the cells before intervention. Thus, we examined the effect of treating cells with dTAG^V^-1 in the presence of cycloheximide, which inhibits new protein synthesis (Figure 6A). In cells co-expressing *m*FKBP-tagged B8 and GFP-LC/A1, dTAG^V^-1 accelerated the degradation of GFP-LC/A1, whereas dTAG^V^-1-NEG, an inert PROTAC lacking the *m*FKBP-binding ligand, had no effect. The effects of the PROTAC on LC/A1 appear to be more substantial than on *m*FKBP-B8, likely due to the much lower expression levels of LC/A1. The specificity of the effect of dTAG^V^-1 is further confirmed by the observation that GFP-LC/A1 levels were not reduced in cells co-expressing VHH B8 without a *m*FKBP tag (Figure 6A, lanes 5–8). Similar results were observed in M17 neuroblastoma cells (Figure 6B).

As recombinant toxins with better efficacy or novel substrate specificity become available for clinical testing, a significant issue remains whether these novel toxins will present unexpected risks. Our findings suggest that the generation of toxins that have a PROTAC binding domain would provide a safety measure to eliminate the recombinant toxin rapidly on-demand. More generally, this strategy should allow the design of novel recombinant toxins with the desired binding and catalytic activity as well as persistence that can be tuned through proper dosing of the PROTAC.

In this study, we also present proof of principle that readily available PROTACs can be used to modulate the stability of recombinant BoNT/A LC engineered with a targeting domain. We have demonstrated that LC/As fused with a relatively large tag such as GFP are enzymatically active in cells [21,22,25]. However, additional studies will be needed to determine the optimal targeting domain to ensure activity of recombinant holotoxins in vivo. Alternatively, the targeting domain can be engineered on a camelid antibody that specifically recognizes BoNT/A LC. Compared to our previous designer E3s [22,25], these approaches allow for on-demand control of the rate of BoNT/A LC degradation. The use of PROTACs to initiate toxin degradation allows not only more precise control of timing, but also permits fine tuning of the rate of toxin degradation by titration of the PROTACs. Since VHHs specific to the light chains of other BoNT serotypes have been developed [26,45,46,47], these can be similarly adapted to target LCs of other BoNT serotypes. This modular design also allows for the use of multiple VHHs or multiple targeting domains to simultaneously target different toxins. The small size of the *m*FKBP tag should permit the delivery of the *m*FKBP-VHHs in vivo using the atoxic BoNT/C1 delivery system [18,44]. In addition, the modular design allows swapping of other small targeting domains as improved PROTACs become available. Additional studies will be needed to determine the optimal in vivo E3 ligand for these PROTACs. VHH-based PROTACs could be useful for shortening the duration of treatment for botulism. Finally, our findings that BoNT LC persistence can be regulated by PROTAC treatment opens the door for the potential development of PROTACs that directly target and ubiquitinate wild-type BoNTs.

## 3. Methods

### 3.1. Plasmids

Construction of recombinant GFP-tagged LC/A1 and LC/A2 has been described [21,22]. GFP-LC/A3 consists of LC/A3 aa 1–445 fused to the C-terminus of GFP and was a gift from George Oyler (Precision Brain MD, Catonsville, MD, USA). To generate Halo-tagged GFP-LC/A1, HaloTag was generated by Polymerase Chain Reaction (PCR) using pHTN HaloTag CMV-Neo (Promega, Madison, WI, USA) as a template and subcloned into GFP-LC/A1 at BglII site. VHH B8 with a C-terminal E-tag was generated by PCR Trx/ALc-B8/E as template and cloned into pcDNA3.1(+) between BamHI and XhoI sites. *m*FKBP(F36V) was synthesized by IDT (gBlock) and subcloned into pcDNA3.1(+) between HindIII and BamHI sites; VHH B8 and LC/A1 were subcloned into the resulting vector between BamHI and XhoI sites.

### 3.2. Antibodies

Antibodies used: Monoclonal antibody against BoNT/A light chain (BEI), GFP, and ubiquitin (Santa Cruz); rabbit polyclonal antibodies against red fluorescent protein (RFP; Abcam, Cambridge, UK), GFP (Clontech, Mountain View, CA, USA), and E-tag (Cell Signaling, Danvers, MA, USA). 

### 3.3. PROTACs

PROTACs were purchased from commercial sources: dTAG^V^-1, dTAG-13, and dTAG^V^-1-NEG (Tocris Bioscience, Bristol, UK); HaloPROTAC3 (Promega, Madison, WI, USA).

### 3.4. Cell Culture

Cells were maintained in complete media in in a humidified incubator at 37 °C and 5% CO_2_. HEK-293 cells were maintained in DMEM supplemented with 10% (*v*/*v*) fetal bovine serum (FBS), 100 units/mL penicillin, 100 μg/mL streptomycin, and 2 mM glutamine. BE(2)-M17 human neuroblastoma cells (ATCC) were maintained in DMEM media containing 10% FBS, 0.1 mM β-mercaptoethanol, 2 mM glutamine, 100 units/mL penicillin, 100 μg/mL streptomycin, and 2 mM nonessential amino acids (NEAA). 

### 3.5. Ubiquitination Experiments

Cells were transfected with indicated plasmids using PolyFect (Qiagen, Hong Kong) according to the manufacturer’s recommendation. Experiments were carried out 36 h post-transfection. Cells were treated with HaloPROTAC3 (20 μM) or vehicle for 20 h, followed with MG132 (30 μM) for 4 h to accumulate ubiquitinated proteins. Cells were lysed in 1% SDS and diluted with PBS pH 7.4 containing 1% Triton X-100, cOmplete Protease Inhibitor Cocktail EDTA-free (Roche, Basel, Switzerland), and 30 μM MG132 (APExBio, Houston, TX, USA) and 10 mM iodoacetamide (Sigma, Kawasaki, Japan). Insoluble materials were removed by centrifugation at 5000× *g* for 5 min and the clarified lysates were precleared by rotating for 1 h at 4 °C with Protein G agarose (Sigma). Immunoprecipitation was performed by incubating the precleared lysate for 4 h at 4 °C with GFP antibody (Clontech) and Protein G agarose on a rotating platform. The immune complexes were washed three times with 20 bed volumes of PHS pH 7.4, 0.5% Triton X-100, and processed for SDS-PAGE and immunoblotting (IB).

### 3.6. Cell Culture Experiments

Cells were transfected with indicated plasmids using PolyFect (Qiagen) or Lipofectamine 3000 (Thermo Fisher, Waltham, MA, USA) according to the manufacturer’s recommendation. Experiments were carried out 36 h post-transfection. For cycloheximide chase, cells were treated with 50 μg/mL cycloheximide for the indicated times. For DUB inhibitors, cells were treated with cycloheximide and either DMSO (solvent; CTL, Beaverton, OR, USA), 2 μM WP1130, or 10 μM P22077. For PROTAC experiments, cells were treated for 24 h. Serial 1:2 serial dilutions of HaloPROTAC3 starting at 40–50 μM were used. With dTAG^V^-1, serial 1:5 dilutions starting at 10 μM were used. When used in combination with cycloheximide (10 μg/mL), dTAG^V^-1 or dTAG^V^-1-NEG was added to a final concentration of 200 nM. Cells were treated for 16 h before lysis in PBS pH 7.4, 1% Triton-X-100, 0.5% sodium deoxycholate supplemented with 1 mM EDTA, cOmplete Protease Inhibitor Cocktail EDTA-free, and 30 μM MG132. Insoluble materials were removed by centrifugation at 5000× *g* for 5 min and the clarified lysates were denatured by heating in reducing LDS sample buffer (Thermo Fisher) at 70 °C for 10 min, resolved by SDS-PAGE, and processed for immunoblotting. Immunoblots were developed using primary antibodies followed by HRP-conjugated secondary antibodies. Images were acquired using an Azure Biosystems c280 imager. All experiments were performed in triplicates. Representative results are shown in figures.

## Figures and Tables

**Figure 1 ijms-25-07472-f001:**
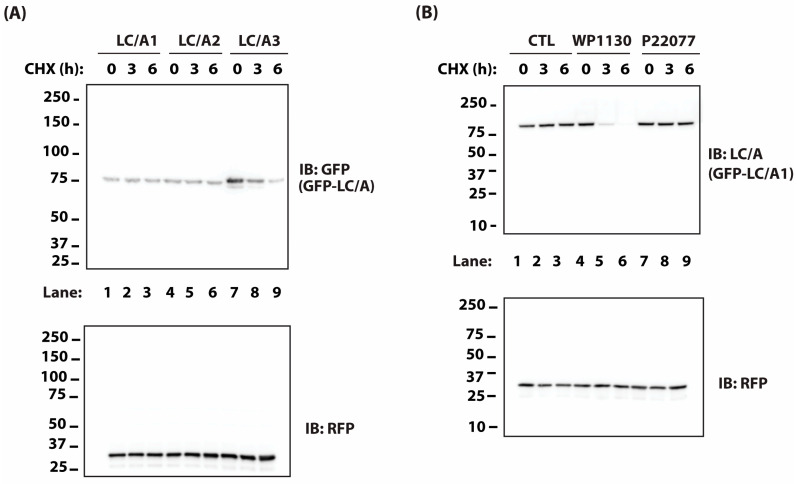
(**A**) Cells were transfected with plasmids encoding GFP-tagged LC/A1 (Lanes 1–3), LC/A2 (Lanes 4–6), or LC/A3 (Lanes 7–9). After 36 h, cells were treated with 50 μg/mL cycloheximide (CHX) for the indicated times and levels of GFP-LC/As were assessed by immunoblotting (IB) with GFP antibody. (**B**) Cells transfected with GFP-LC/A1 were treated with CHX and DMSO (Lanes 1–3), 2 μM WP1130 (Lanes 4–6), or 10 μM P22077 (Lanes 7–9). Levels of GFP-LC/A1 were assessed by IB. RFP serves as a transfection efficiency control in (**A**,**B**).

**Figure 2 ijms-25-07472-f002:**
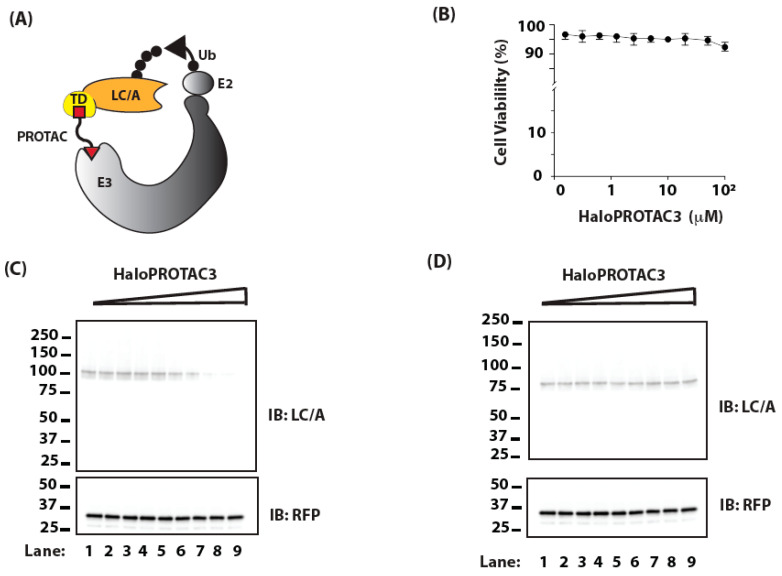
(**A**) Schematic representation of how a cellular E3 can be redirected to target LC/A using a PROTAC designed to bind a specific targeting domain (TD). The PROTAC binds the TD on one end and recruits the E3 ligase complex that binds to a ubiquitin-conjugating enzyme (E2) charged with ubiquitin (Ub) on the other. (**B**) Cells were treated with increasing doses of HaloPROTAC3 for 24 h and cell viability was assessed with ethidium homodimer-2. Data shown are mean ± SD (n = 3). (**C**) Cells transfected with plasmids encoding GFP-Halo-LC/A1 were treated with vehicle (Lane 1) or increasing concentrations of HaloPROTAC3, ranging from ~390 nM (Lane 2) to 50 μM (Lane 9), for 24 h. Levels of GFP-Halo-LC/A1 (arrow) were assessed by IB. (**D**) Cells transfected with plasmids encoding GFP-LC/A1 lacking the HaloTag were treated with vehicle (Lane 1) or increasing concentrations of HaloPROTAC3, ranging from ~390 nM (Lane 2) to 50 μM (Lane 9), for 24 h. Levels of GFP-LC/A1 were assessed by IB; RFP served as a transfection control in (**C**,**D**). (**E**) Dose responses for GFP-Halo-LC/A1 and GFP-LC/A1 are from data in (**C**,**D**). (**F**) Cells were transfected with plasmids encoding GFP-Halo-LC/A1 and treated for 20 h with 20 μM HaloPROTAC3, followed by 4 h with 30 μM MG132. Cells were lysed in denaturing buffer and processed for immunoprecipitation of GFP-Halo-LC/A1. Ubiquitin and LC/A1 were detected by IB. The arrow indicates the expected apparent molecular weight of unmodified GFP-Halo-LC/A1.

**Figure 3 ijms-25-07472-f003:**
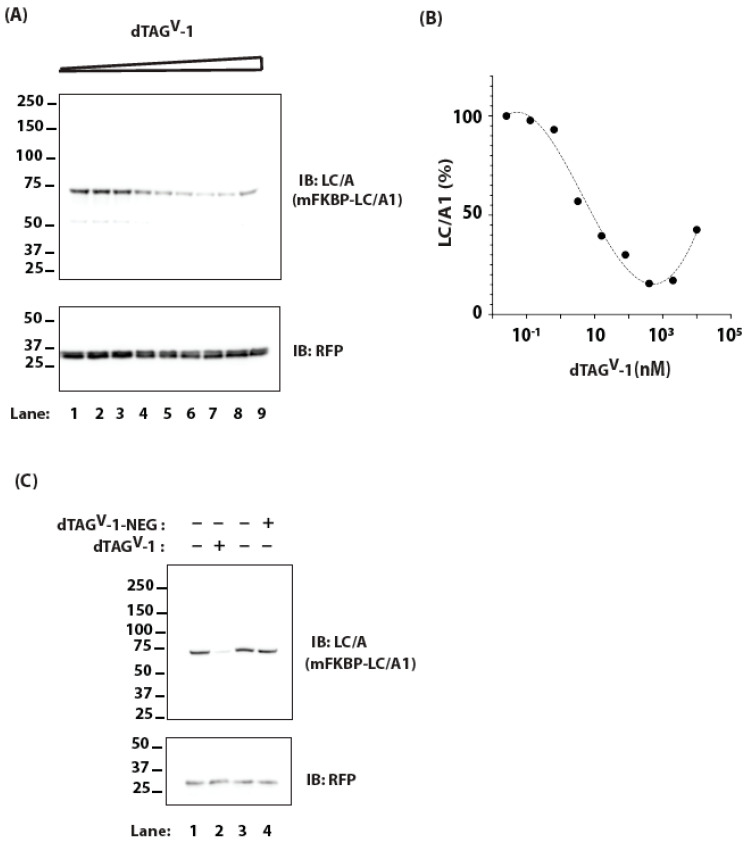
(**A**) Cells transfected with *m*FKBP-LC/A1 were treated with increasing concentrations of dTAG^V^-1, ranging from ~0.1 nM (Lane 1) to 10 μM (Lane 9), for 24 h. Levels of *m*FKBP-LC/A1 were monitored by IB. RFP serves as a transfection efficiency control. (**B**) Dose response for *m*FKBP-LC/A1 is derived from data in (**A**). (**C**) M17 neuroblastoma cells transfected with *m*FKBP-LC/A1 were treated with 10 μg/mL CHX for 16 h in the presence of 200 nM dTAG^V^-1 or dTAG^V^-1-NEG (inert PROTAC, negative control). Levels of *m*FKBP-LC/A1 were assessed by IB. RFP serves as a transfection efficiency control. Levels of *m*FKBP-LC/A1 were monitored by IB. RFP serves as a transfection efficiency control.

**Figure 4 ijms-25-07472-f004:**
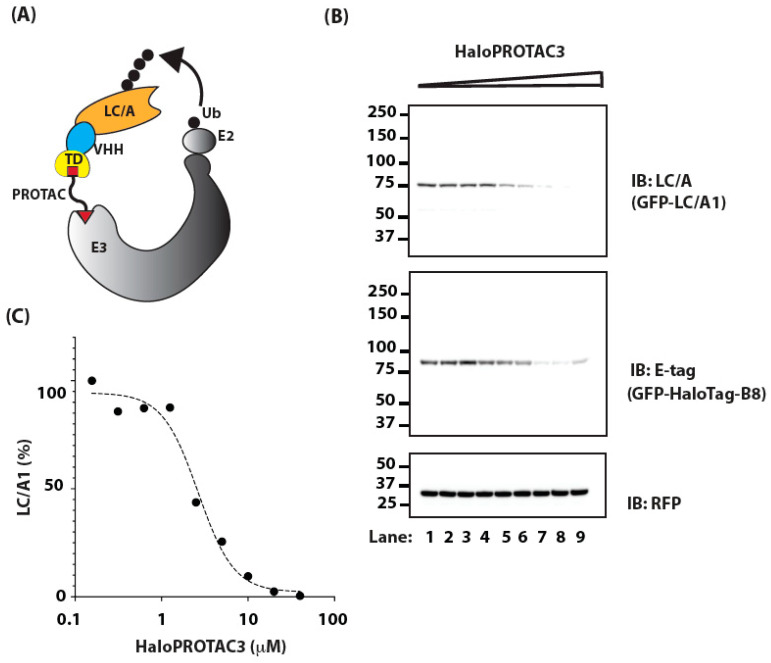
(**A**) Schematic representation of targeting LC/A with PROTAC using an intermediary protein. Here, an LC/A-specific VHH is fused to the targeting domain (TD) and serves to couple the PROTAC to the LC/A and E3 complex. (**B**) Cells transfected with plasmids encoding GFP-Halo-VHH and GFP-LC/A1 were treated with increasing concentrations of HaloPROTAC3, ranging from ~156 nM (Lane 1) to 40 μM (Lane 9), for 24 h. Levels of GFP-LC/A1 and GFP-Halo-VHH were assessed by IB. RFP serves as a transfection efficiency control. (**C**) Dose response for GFP-LC/A1 is derived from data in (**B**).

**Figure 5 ijms-25-07472-f005:**
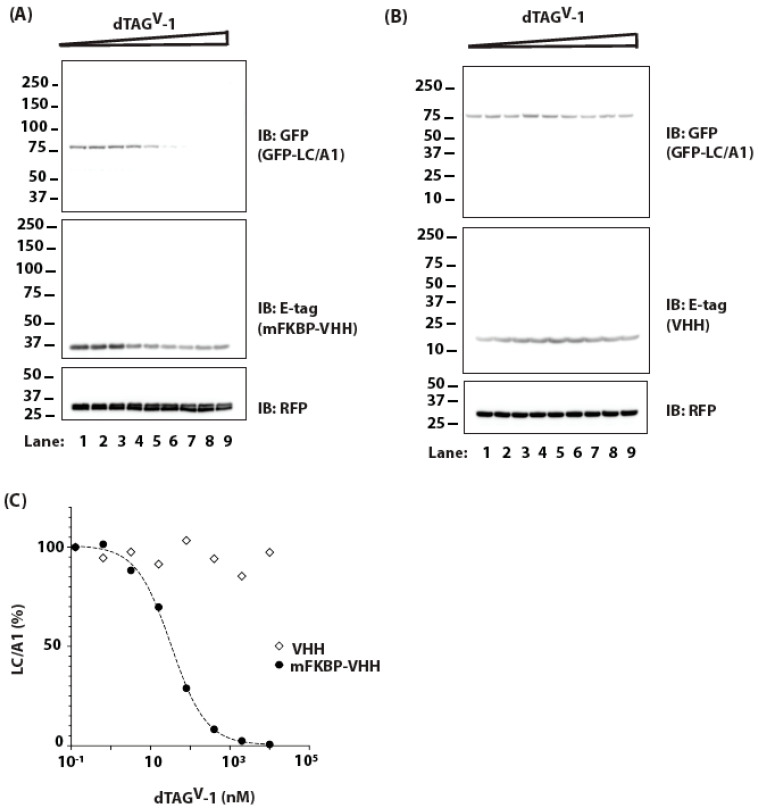
(**A**) Cells transfected with plasmids encoding *m*FKBP-tagged VHH and GFP-LC/A1 were treated with increasing concentrations of dTAG^V^-1, ranging from ~0.1 nM (Lane 1) to 10 μM (Lane 9), for 24 h. Levels of GFP-LC/A1 and *m*FKBP-tagged VHH were assessed by IB. (**B**) Cells transfected with plasmids encoding VHH B8 and GFP-LC/A1 were treated with increasing concentrations of dTAG^V^-1, ranging from ~0.1 nM (Lane 1) to 10 μM (Lane 9), for 24 h. Levels of GFP-LC/A1 and VHH B8 were assessed by IB. RFP serves as a transfection efficiency control in (**A**,**B**). (**C**) Dose response for GFP-LC/A1 in the presence of B8 vs. *m*FKBP-B8 from (**A**,**B**).

**Figure 6 ijms-25-07472-f006:**
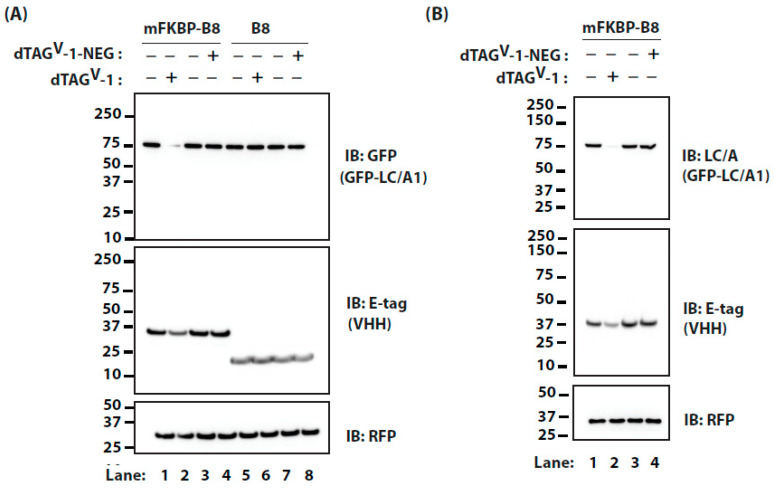
(**A**) Cells were transfected with plasmids encoding GFP-LC/A1 and either VHH (Lanes 5–8) or *m*FKBP-tagged VHH (Lanes 1–4). After 40 h, cells were treated with 10 μg/mL CHX for 16 h in the presence of 200 nM dTAG^V^-1 or dTAG^V^-1-NEG (inert PROTAC, negative control). Levels of GFP-LC/A1 and VHH were assessed by IB. (**B**) M17 neuroblastoma cells were transfected with plasmids encoding GFP-LC/A1 and *m*FKBP-tagged VHH. After 40 h, cells were treated with 10 μg/mL CHX for 16 h in the presence of 200 nM dTAG^V^-1 or dTAG^V^-1-NEG (inert PROTAC, negative control). Levels of GFP-LC/A1 and VHH were assessed by IB. RFP serves as a transfection efficiency control in (**A**,**B**).

## Data Availability

Data are contained within the article.

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
