# Peer review of "The Degradation of Botulinum Neurotoxin Light Chains Using PROTACs"

_ijms, 2024, doi:10.3390/ijms25137472_

Round 1

Reviewer 1 Report

Comments and Suggestions for Authors

The manuscript shows data on the use of PROTACs to accelerate the degradation of the BonNT toxin and proposes this strategy as a therapeutic alternative for the treatment of botulism. The results are interesting and clearly demonstrate that the different PROTACs constructs are effective in promoting the degradation of the toxic LC/A1 fraction. However, I have two concerns that need to be addressed.

1.     In my opinion, it is somewhat confusing that the authors evaluate the degradation of GFP-LC/A1 using either anti-GFP or anti-LC/A1 antibodies. To present solid results and demonstrate that the LC/A1 protein is degraded, the anti-LC/A1 antibody should be consistently used to directly analyze LC/A1-specific proteolysis. Otherwise, using the anti-GFP antibody, it is not clear whether LC/A1 remains intact.

2.     The authors propose that PROTAC-mediated degradation occurs via ubiquitination of LC/A1. Therefore, it is crucial to analyze whether LC/A1 is ubiquitinated in the presence of different PROTAC constructs.

3.     Figure 2B shows several protein bands with lower molecular weight, whose degradation is consistent with the degradation of LC/A1. The authors should clarify the nature of these bands. Does their degradation also depend on PROTAC-mediated ubiquitination? If so, does this mean that the PROTAC system is nonspecific?

Reviewer 2 Report

Comments and Suggestions for Authors The paper describes that the use of two different approaches to tune the levels of botulinum neurotoxins light chains by selectively targeting their catalytic light chains for proteasomal degradation. This title “Tunable Persistence of Botulinum Neurotoxin using PROTACs” is not very suitable, which may be revised into “The degradation of Botulinum Neurotoxin light chains using PROTACs”. This title should be more suitable. In addition, this work had better do or test it in neuro cells to further support it. In Figure 4,GFP-Halo-tagged-B8 is incorrect, it is Halo-tagged-B8, without GFP. Overall, the paper is interesting and important.

Reviewer 3 Report

Comments and Suggestions for Authors

Botulinum neurotoxin (BoNT) is composed of a cell-binding heavy chain and a catalytic light chain (LC) that enters the host cytosol to cleave a SNARE protein, thereby inhibiting neurotransmission.  This paper describes the use of proteolysis targeting chimeras (PROTACs) as a strategy to modulate the cytosolic persistence and, thus, potency, of the BoNT LC.  Two variations of the strategy were employed, involving either direct PROTAC targeting to a recombinant BoNT LC or indirect PROTAC recruitment to the native BoNT LC.  The former strategy could be used to fine-tune the persistence of clinically applied BoNT, while the latter approach could be used as a treatment for botulism.  PROTAC-directed toxin degradation was monitored in cultured cells transfected with plasmids encoding variants of the BoNT LC.  The work is straightforward and clearly presented but could be improved by acknowledging the limitations of the experimental approach, clarifying the protocols, and correcting a few overstatements.

Limitations

1.  All studies in this work were conducted with transfected, non-neuronal HEK-293 cells that directly express a variant of BoNT LC in the cytosol.  No experiments with recombinant holotoxins and neuronal cells were performed.  Thus, it is unknown whether (i) the recombinant tags added to the LC would interfere with holotoxin assembly, (ii) the variant LCs can reach the cytosol when applied exogenously as part of a BoNT holotoxin (for clinical applications), and (ii) how the tags affect toxin potency.  Given the emphasis on potential clinical applications of this work, these limitations should be addressed in the manuscript.

2.  An 8 uM concentration of HaloPROTAC3 was required to achieve half-maximal degradation of HaloTag-LC.  This seems to be a very high concentration for potential therapeutic application, and HaloPROTAC3 may be toxic at concentrations above 10 uM.  This paper is a proof-of-concept study, but the authors should still consider how their system would need to be optimized for an actual treatment.  

Clarifications

1.  Why do the transfected BoNT LC constructs contain a GFP tag?  GFP is very stable and could potentially alter the kinetics of LC degradation.  The authors have an antibody against LC, so why did they blot against the GFP tag instead of directly monitoring the turnover of LC with the anti-LC antibody?

2.  Why does Figure 1 not include an RFP blot as a control for transfection efficiency?

3.  It appears that there is some loss of the untagged LC control at high HaloPROTAC3 concentrations (Fig. 2C-D).  The authors should accordingly revise their statement that “HaloPROTAC3 had no effect in the absence of HaloTag” (line 162).  The graph may underestimate the extent of untagged LC degradation, as it appears the condition set as 100% protein was underloaded compared to the other samples (ie, several points on the graph are above 100%).  Is the PROTAC actually promoting degradation of untagged LC, or is the PROTAC potentially toxic at high concentrations?  The authors note that HaloPROTAC3 is not cytotoxic up to 10 uM, but three of their tested concentrations are at or above 10 uM and could therefore have an indirect effect on LC stability through cytotoxic effects.  This should be addressed in the text.

4.  In Figure 2, the blots show results using 9 different PROTAC concentrations.  The graph only shows six data points.  Why does the graph not plot the results from all 9 PROTAC concentrations?

Overstatements

1.  The abstract states that BoNTs are the most potent natural toxins known, but tetanus toxin is generally considered to be as potent as BoNT.  The authors should acknowledge this or simply rephrase the sentence to read “BoNTs are some of the most potent…”.

2.  The authors state in the Introduction that “there is currently no clinical treatment for botulism” (line 57), but BoNT antitoxin is a standard treatment for botulism.  The authors acknowledge this at the end of the paper (lines 236-237).

Minor points

1.  The PROTAC abbreviation should be defined in the title and abstract.

2.  Lines 186-187:  This sentence appears to state that the LC construct reached a low concentration of 1 mM, but the concentration of the LC was not determined.  The sentence should be rewritten to clarify that (I think) the 1 mM concentration is in reference to dTAG:  the greatest extent of LC degradation was obtained with this concentration of dTAG.  Also, is it 1 mM or 1 uM?

3.  Line 309:  “FOR” should be “For”.

Round 2

Reviewer 1 Report

Comments and Suggestions for Authors

The authors have addressed my concerns and I am satisfied with the response. I recommend the publication of the manuscript

Reviewer 2 Report

Comments and Suggestions for Authors The revision got a promotion and solved my problem.